# Management practices in hospitals: A public-private comparison

**Claudio Lucifora** *

Department of Economics and Finance, CRILDA and IZA, Università Cattolica del Sacro Cuore, Milano, Italy

* claudio.lucifora@unicatt.it

## Abstract

We use information on management practices in 1,183 hospitals in 7 different countries, collected in 2010 within the "World Management Survey" initiative, to estimate the role of public ownership on different management dimensions, such as monitoring performance, setting targets and incentivizing employees. A significant variation in management practices both between countries and, within countries, across hospitals is found. We show that managers in public sector hospitals tend to underperform, relative to private hospitals, in all the countries considered. Larger hospitals appear to be better managed, while there is no difference between teaching and other type of hospitals. Publicly owned hospitals appear less efficient in the provision of incentive schemes to promote and reward highly motivated employees, or remove poor performers. Overall, public ownership is associated with a reduction of about 10% in management score, which corresponds approximately to a half-standard deviation.

## Introduction

The COVID-19 pandemic has put the healthcare sector and its functioning back at the center stage of the political debate. The resilience of countries in responding to the health consequences of the COVID-19 pandemic crisis has been quite differentiated both in terms of countries' ability to re-organize and activate healthcare systems. Coordination mechanisms proved to be very important to prevent escalation and control damage, particularly in those countries where regions and municipalities carry responsibility for public health services and hospital spending, as resources have to be mobilized at the national level to face the heterogeneous distribution of COVID-19 outbreaks. In this context, the capacity and efficiency of hospitals to provide emergency support, to reorganize and open new intensive care units (ICU) has been crucially dependent on managers and managerial practices in hospitals [1]. While differences in hospital efficiency depend on a wide range of factors, which are generally difficult to measure, a number of recent studies have shown that management practices can be an important driver of such differences [2]. Much of this research has focused on cross-country differences in management practices across organizations in manufacturing and retail industries, as well as education and healthcare. Another dimension of differences in performance and management practices is ownership. Bloom et al [3], for example, show that firms owned by family members are generally badly managed compared to firms run by professional CEO. In general, available evidence shows that public sector organizations tend to be characterized by worse

have been uploaded as Supplementary Information.

**Funding:** C.L. gratefully acknowledges financial support from Fondazione ROdolfo DeBenedetti (fRDB) (https://www.frdb.org/). The funders had no role in study design, data collection and analysis, decision to publish, or preparation of the manuscript.

**Competing interests:** The author declares that no competing interests exist.

management practices, even after controlling for a number of compositional factors. The presence of regulatory constraints, low power incentives, strong unions and limited competition are often cited in the literature to explain the lower performance of government-owned organizations [4].

In this paper, we focus attention on the public-private differences in management practices in healthcare sector and specifically in hospitals. We build on the work of Bloom and Van Reenen [5] and use data collected in 2010 within the "World Management Survey" initiative (WMS) which cover more than 8,000 public and private organizations in 20 different countries. In the empirical analysis, we focus on the sub-sample of hospitals in five European countries, Canada and the US. More details on the structure of the interviews and the methodology used to quantify management practices are given in the data section below.

Indeed, hospitals behavior is an interesting case to study for a number of reasons. First, as discussed above, the recent virus epidemic has shown that hospitals represent the ultimate safety net to provide health assistance to the population at large and, in emergency situation, only good management practices can guarantee an efficient provision of health services to unanticipated re-organization needs (e.g. recall retired physicians, transfer patients to other hospitals, convert hospital departments into ICU, etc.). Second, in recent years, hospitals have experienced significant financial pressure to improve quality standards, while facing cuts in their financing which increased heterogeneity across areas, particularly in federal states. Third, hospitals can differ along several dimensions, such as the type of care they provide, their size, the ownership structure and whether they are research or teaching hospitals, also even when they offer similar services across different countries, the healthcare institutional setting may differ significantly. Fourth, while public and private hospitals coexist and, to some extent, compete on the market, they tend to have different institutional goals, with private hospitals often oriented to profit and shareholder value, while public hospital are expected to pursue the public interest, and directly or indirectly, be more accountable to politics [6]. Some of the features that characterize the internal functioning of hospitals suggest that the structure of incentives confronting management strategies may be similar, while external factors such as: government regulations in recruitment and compensation policies, presence of unions, and institutional barriers to competition, just to list a few, may play a larger role in public sector hospitals. Typically, public hospitals have a larger board and they are more likely to have politicians in it, which can often worsen financial performance [2].

It is important to stress that what is under investigation here is not the quality of healthcare or the services provided to patients, nor the resilience of hospitals to the coronavirus outbreak, but simply the management practices adopted in public and private hospitals, during normal times, to monitor performance, set targets, and recruit, retain and motivate the personnel. While these management practices are certainly not the only dimensions relevant for the performance of private and public organizations, the selected dimensions effectively synthetize a larger pool of simple indicators collected in the interviews, as described below (see S1 Table in S1 Appendix), which have been shown to play a key role in management practices across different type of organizations [2].

The paper is organized as follows. First, we review the recent literature on management practices with a focus on hospitals. Second, we describe the methodology used to measure management practices, we describe the data, and we outline the empirical strategy to estimate the effects of public ownership. Third, we present the main set of results, both in terms of empirical regularities, as well as estimating the management score gap between public and private hospitals. We also discuss the limitation of the analysis and the role of hospital competition on management standards. Finally, we review the main contribution of the study and their implications for hospital public ownership.

## A review of the literature

### Management practices

In this section, we provide a brief overview of the evolution of managerial practices in the public sector from the Weberian model of bureaucracy to the so-called New Public Management (NPM) and discuss the recent push toward relational public management [7]. Managerial practices in the public sector are traditionally described as strongly hierarchical, governed by formal procedures intended to restrict bureaucrats' discretion and prevent corruption. In such context, civil servants adhere to a set of practices which are uniform across most of the public sector, while employees enjoy lifelong careers and strong job security [8]. During the 1990s public organizations went through a deep transformation known as NPM intended to align managerial practices to those employed in the private sector so as to improve efficiency and reduce costs [4,9]. The main features of NPM were: (i) strengthening of management functions, (ii) changes in the organizational structure of government agencies, and (iii) stronger orientation to the market. Under NPM, public sector managers were given greater discretionary powers and tighter control over personnel through performance targets and appraisals. In exchange, they were expected to improve organizational performance. A process of decentralization also followed, whereby large centralized government agencies evolved into smaller and partially independent units that were devolved significant managerial responsibilities. Finally, competitive tendering and internal markets were used to increase competition across units belonging to the same organization. Although the principles of NPM rapidly diffused across most OECD countries, a few decades after their implementation the feeling is that the expected change fell short of expectations [10]. For a critique to NPM see [11]. A systematic review of the recent studies on managerial practices in public sector organizations is provided in [7], where they argue that the NPM approach to people management has overstated formal management tools and financial incentives rather than try to leverage a broader range of motivations and build organizational culture.

Several studies in the area of managerial practices have focused on strategic human resources management. The main findings show the link between the organization's business strategy and its human resource strategy, covering issues such as training of managers, managerial satisfaction, compensation and motivation [12]. Neelankavil, et al [13] examine differences in managerial performance of middle-level managers in four countries (China, India, the Philippines, and the United States) finding important differences in factors affecting managerial performance as perceived by the respondents–particularly along the East-West dimension.

**Healthcare.** Particularly in the healthcare sector, medical doctors have increasingly been exposed to management control measures with implications for their professional autonomy and control. Numerato et al. [14] compare doctors' perceptions about management practices in two Italian regions (Lombardy and Emilia-Romagna) which differ in the values of the political environment. A total 220 doctors working in public hospitals were surveyed, asking them to report their (perceived) professional freedom. In Emilia-Romagna doctors perceived their organization to be more managerially driven, while in Lombardy doctors reported higher professional freedom thus suggesting that local values may have tangible effects on hospital management. Fattore et al. [15] provide a comprehensive overview of the literature dealing with the relationship between clinicians and management practices arguing that the emphasis on task-related dimensions of professionalism, and the hegemony-resistance framework that is prevalent in most current analyses, has limited the scope of the analysis of the impact of management practices in the medical profession, to include also nurses and healthcare managers. In general, more recent studies tend to emphasize the importance of the relationship between

workers and the network they operate in, as well as with organizational culture. To the extent that these differ between organization operating in the public and private sector, such as public and private hospitals, we expect to find differences in management practices and ultimately on their performance [7].

Another strand of the literature has focused on managerial practices as a way to explain the large productivity differentials that are observed across organizations [16]. In their seminal paper, Bloom and Van Reenen [5] developed an innovative methodology to measure management practices. While the interviews are generally conducted with employees and middle managers, and as such are more likely to record perception of management styles rather than actual managerial practices as designed by top managers, the double-blind methodology used in the survey to code the responses is shown to be robust to measurement error [16,17]. Bloom et al. [2] analyzed management practices in a large number of hospitals in the UK, and found that "best practices" were associated with better patient outcomes–i.e. hospital length of stay and risk-adjusted mortality rate. Chandra et al. [18] found a positive association between measured management scores in US hospitals and acute myocardial infarction. A positive relationship between patient outcomes, hospitals' management standards and proximity to medical schools has also been shown in Bloom et al [19]. Baker et al. [20] investigated hospital ownership in relation to a number of system operations, such as financial aspects, management practices and personnel issues. They show that organizational outcomes differ among hospital ownership types, but in general the evidence on management performance, and patients' outcomes is mixed or inconclusive. Differences in ownership types are minimized in more competitive environments. Other studies have relied on readily available survey data to construct alternative measures of management practices in the public sector [21,22].

## Methodology

### Measurement of management practices

The information on management practices we use in this section relies on the data collected by Bloom and Van Reenen [5] within the WMS initiative (https://worldmanagementsurvey.org/). Below, we provide the main features of the approach used in the WMS, while we refer to Bloom and Van Reenen [2] and to S1 Table in S1 Appendix for further details. Management practices were collected in telephone interviews with clinical service leads (in cardiology and orthopedics units) in acute care hospitals, who were asked about a number of key day-to-day operations. Interviews were conducted using a "double-blind" technique: individuals were not told they were being scored, and interviewers did not know anything beforehand about the hospital they surveyed. The interviewers asked open questions on a checklist of 18 management practices and then scored each practice using a grid ranging from 1 ("worst practice") to 5 ("best practice"). The 18 practices refer to 3 broad areas: performance monitoring, target setting and incentives (see Table A1 in the online Appendix). The scores provide a metric to identify and measure the best practices adopted in surveyed hospitals. In particular, *monitoring* is intended to assess how hospitals monitor operations inside the various units and whether this information is used for continuous improvement; *targets* refers to the presence of objectives, the choice of outcomes and what action is generally taken when the two diverge; *incentives* concerns how employees are hired, retained and whether their performance is rewarded. The overall measure of management practices is computed averaging scores out of the 18 practices recorded. In this context, a low score is associated with a "bad practice", that is a hospital with no performance monitoring, no target setting, and no rewards based employees' performance. Conversely, a "good practice" identifies a hospital with intensive monitoring of performance, well identified targets, and supporting performance-related-incentives. These criteria refer to standard

management practices, as traditionally perceived and shared by HR professionals, around the world. Finally, in order to address concerns about the self-assessed nature of managers' answers and the relationship with actual performance, WMS data has been validated through a number of consistency checks, both 'internal' (i.e. cross-validation, double-scoring, re-surveys) as well as 'external' (i.e. "best practices" shown to be highly correlated with various performance indicators). While correlations provide suggestive evidence that management practices are associated with organizational performance, the non-experimental setting of most studies raises concerns that such relationship might not be causal. In other words, reverse causality—as better managers and better practices are more likely to be adopted in high performing hospitals—, and unobservable factors—such as environmental attributes, organization climate, personnel motivation, etc. —, may confound the relationship between management and performance. The present study shares most of the caveats with the literature discussed above. In particular, the cross-sectional nature of the WMS data on hospitals and the limited information on hospitals' characteristics severely limit our ability to address the selection between management practices and unobserved hospitals characteristics. However, even if we cannot provide strong causal evidence about the effect of public-private ownership on management practices, still by leveraging on variations across countries, hospital types and institutions (in WMS data), we can describe a number of interesting patterns in management practices across hospitals and countries. A number of checks also support the robustness of the results.

## Data and analysis

In the empirical analysis, we use the hospital sub-sample in the WMS data. Overall, the data provide information on hospital's characteristics (number of beds, ownership and teaching hospital), as well as management practices for 1,183 hospitals, operating in the following countries: United States, Canada, United Kingdom, Sweden, Germany, France and Italy. The majority of the hospitals surveyed has a public ownership (70 percent), though the public-private hospital shares, in the sample, vary substantially across countries: for example, ownership is distributed more evenly in the US, UK and Germany; less so in France, Italy and Canada. The average size of surveyed hospital is close to 350 beds, with average hospital size significantly larger in France, but much smaller in the US. Also public hospitals tend to be generally larger compared to private ones, with varying differences across countries in the number of beds.

In order to empirically validate the stylized fact presented in the previous sections, in this section we specify and estimate a simple linear relationship between management scores and a public-private sector dummy also controlling for a number of hospital attributes and country fixed-effects. Since hospitals can differ along several dimensions, it is important to control for hospital characteristics. For example, larger hospitals are more difficult to be managed compared with smaller ones, hence it is important to control for hospital size. Teaching hospitals tend to be affiliated with a university, medical or nursing schools and conduct academic medical research, conversely non-teaching hospitals mainly serve the local community. Since these features can interact with the complexity of the tasks and the skills of the medical and nursing staff, also controlling for the teaching-nonteaching status of the hospital is important.

In practice, we use information on management practices, hospital's ownership and the number of beds available in the WMS data, pooling all hospitals across different countries, to estimate public-private sector differences in management practices. Our dependent variable is $MS_{ic}^k$, which records the overall score of management practices in hospital $i$ and country $c$, while the superscript $k$ identifies the different management dimensions, such as: 'Performance monitoring', 'Target setting' and 'Incentives'. We define a dummy variable, Public Hospital,

which takes value 1 for public sector hospitals and 0 for private ones. We control for a number of hospitals' attributes ($X$) such as: hospital size, measured by the number of hospital beds (3 dummies, small: less than 50 beds, medium: 50 to 250 beds and large: more than 250 beds, or as the log of the number of beds), and teaching *versus* non-teaching hospital. Finally, since, the survey covers different countries, we also include country fixed-effects ($\theta_c$) to account for country specific characteristics (i.e. institutional settings and healthcare system). In practice, we specify and estimate the following relationship by simple least squares,

$$MS_{ic}^k = \alpha_c + \beta_c Public\ Hospital_{ic} + X\prime_{ic}\delta_c + \theta_c + \varepsilon_{ic} \qquad (1)$$

where $\beta_c$ is our main coefficient of interest. One concern is related with the limited number of hospital-level controls that are available in the WMS dataset. A number of relevant characteristics associated with management practices, such as: the served population, the organizational complexity–which is only partially captured by hospital size–or its overall performance, for which we cannot control, may contribute to omitted variable bias (OVB) and affect the robustness of our results. To address this issue, in the robustness section, we take a number of steps which leverage on the distribution of the unobservables. Furthermore, since the specification of model (1) above implicitly assumes that public ownership is randomly distributed across hospitals, in a number of additional exercises we extend the model to allow for the presence of endogenous selection in hospital ownership. Indeed, as previously discussed, better managers and better practices may be correlated with unobservable factors related with the profitability of the organization. Hence to address this issue, we use a two-step Heckman selection model exploiting information on the extent of competition in the market as instrument for hospital's public ownership. Competition here is defined as the number of hospitals (competitors) that are present in the local market in which the hospital operates, as reported by the respondent. Notice that, while the "intensive" margin of competition–i.e. number of competitors–is likely to be correlated with unobservable characteristics associated with management scores and thus unlikely to satisfy the exclusion restriction, here we mainly exploit the "extensive" margin of competition–i.e. the presence/absence of competitors–which mainly depends on healthcare regulations and institutional features, hence unrelated to management practices (see [3] for a similar approach). In practice, our instrument is defined as a dummy equal to 1 if the hospital has any competitor in the local market.

## Main results

### Descriptive findings

The main descriptive statistics of management scores, for the pooled sample of hospitals, show a significant dispersion, with several hospitals very close to the best practice (score 5), as well as hospitals very poorly managed (score 1). The average management score is slightly lower compared to the value that would result if the quality of practices were randomly distributed across hospitals (score 3) (see S2 Table in S1 Appendix). When considering separate scores for the three broad areas in which practices have been grouped: a) performance monitoring, b) target setting and c) incentives; the area that appears to be better managed is performance monitoring, while the area of incentives shows lower scores. In Fig 1, we report the average score of management practice, separately for public and private hospitals, across countries. In general, private hospitals seem to be better managed (scores are systematically higher), compared to public ones. However, part of the observed dispersion in management scores is likely to reflect differences across countries. Differences in average scores are sizeable: the United States exhibit the highest management score, both in private and public sector hospitals, followed by the UK, Sweden and Canada, while Italy and France are located at the bottom of the

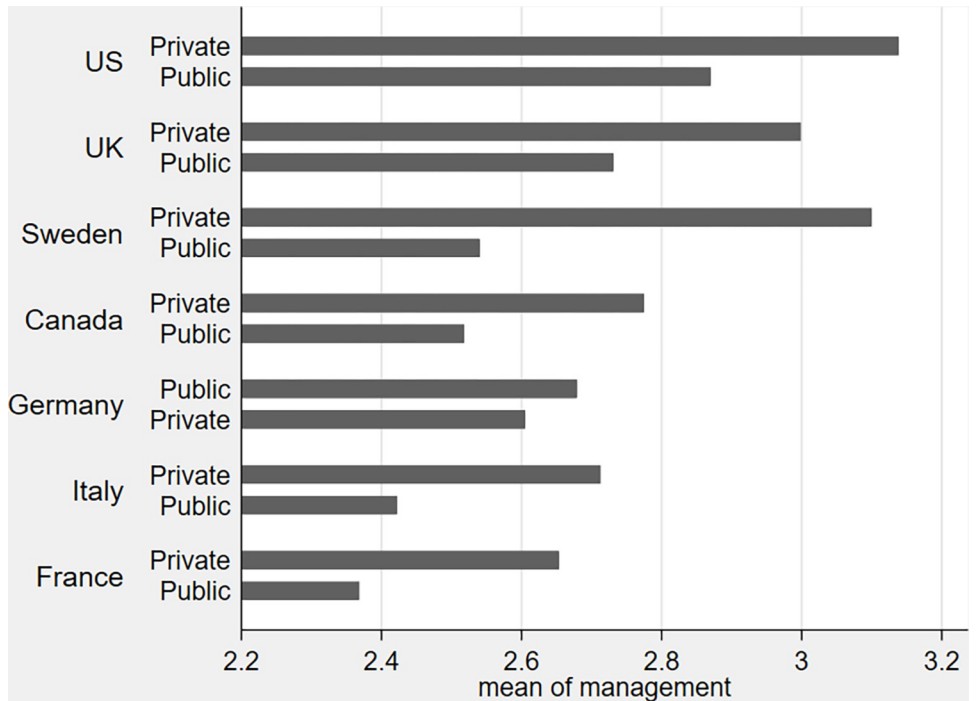

**Fig 1. Public-private average management practices scores (by country).** Source: World Management Survey (Hospitals sub-sample).

ranking. Moreover, in all countries, the (unconditional) average management score of private hospitals always outperform public sector ones. It is interesting to note that the country ranking, reported in Fig 1, closely mirrors observed differences in cross-country overall productivity, thus providing further support to the scoring methodology [23].

Besides country rankings, one additional feature of management practices in public and private hospitals concerns their distribution within country. In other words, we go beyond country averages and investigate the actual distribution of management practices in public and private hospitals. In Fig 2, for each country, we plot a hospital-level histogram of management practices for the public sector and place on top of that a continuous line obtained fitting a kernel function to private sector hospitals (Sweden is excluded due to the scarce number of private hospital surveyed). The variance in management practices scores across hospitals, within country, is considerable and consistent with a management score ratio between 2 and 3, suggesting that hospitals located at the top are two to three times better managed with respect to those at the bottom [24].

Comparing the distribution of management practices in public and private hospitals, we show that continuous lines (private sector) are shifted to the right with respect to the histograms (public sector) confirming previous findings suggesting that private hospitals are better managed than public ones [3]. In particular, two main patterns emerge from Fig 2: the first one, for the US, UK and France, where a larger share of 'best' practices is found in private hospitals; a second one, for Canada and Italy, in which there is a much thicker 'left tail' of badly managed public hospitals. Germany is a special case due to a more compressed distribution of management practices and a bi-modal shape of management density in public hospitals. While the above features reveal interesting insights about how public and private hospitals are managed, nevertheless they hide considerable heterogeneity across countries, and between

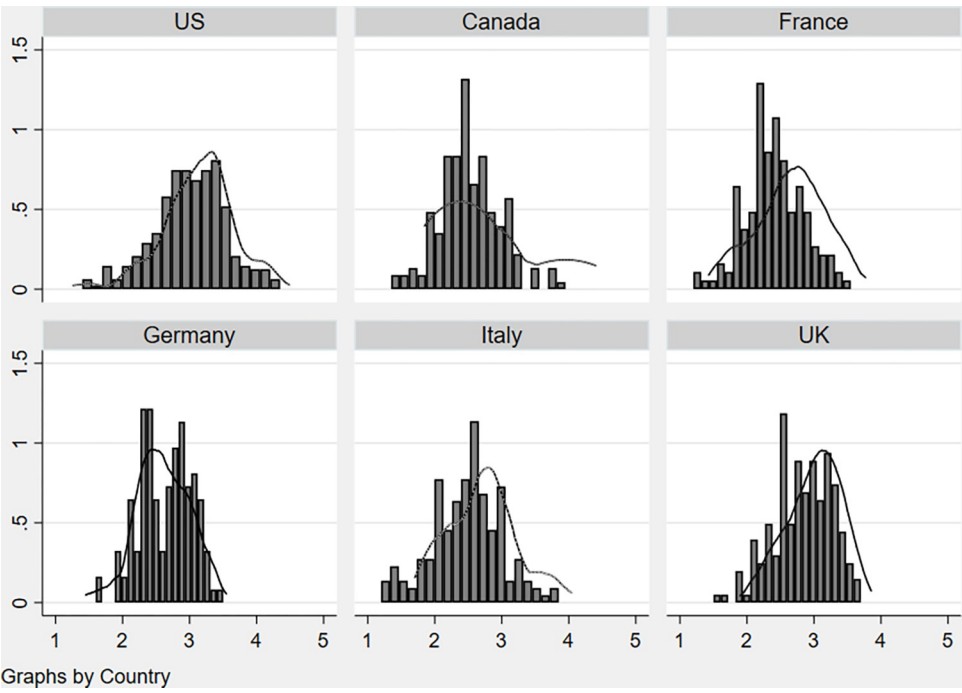

**Fig 2. Distribution of management practices scores (public-private by country).** Source: World Management Survey (Hospitals sub-sample).

public-private sectors, in market structure and hospitals' size. We address some of the above features in the following sections.

## Regression results

In this section we report the main set of results obtained estimating model (1) by Ordinary least squares.

Fig 3 shows coefficient estimates and their confidence intervals (see S3 Table in S1 Appendix). The four different panels refer to the overall 'Management' practices, and then separately for some components of management practices (i.e. 'Performance monitoring', 'Target setting' and 'Incentives', respectively). In line with the descriptive evidence previously reported, public owned hospital show a negative and statistically significant coefficient suggesting a lower overall management score. Evaluated at the mean, public ownership implies a reduction of about 10% in management scores, which corresponds approximately to a half-standard deviation. Comparable results are found with respect to the scores in performance monitoring and target setting. Conversely, the evidence associated with the structure of incentives to recruit, retain, promote and reward high performing employees shows a much larger gap in the public sector–i.e. approximately 30% lower.

Fig 3, also reports the relationship between management scores and both teaching status, and hospital size. The teaching status of an hospital does not seem to be associated with different management practices compared with other hospital. While this result may be unexpected, since teaching hospital generally have a more skilled workforce and are better financed, it should be considered that teaching hospitals tend to treat sicker patients and often run clinical trials, both of which can increase substantially the complexity of management practices making more difficult to obtain a high score. Larger hospitals, conditional on ownership, are

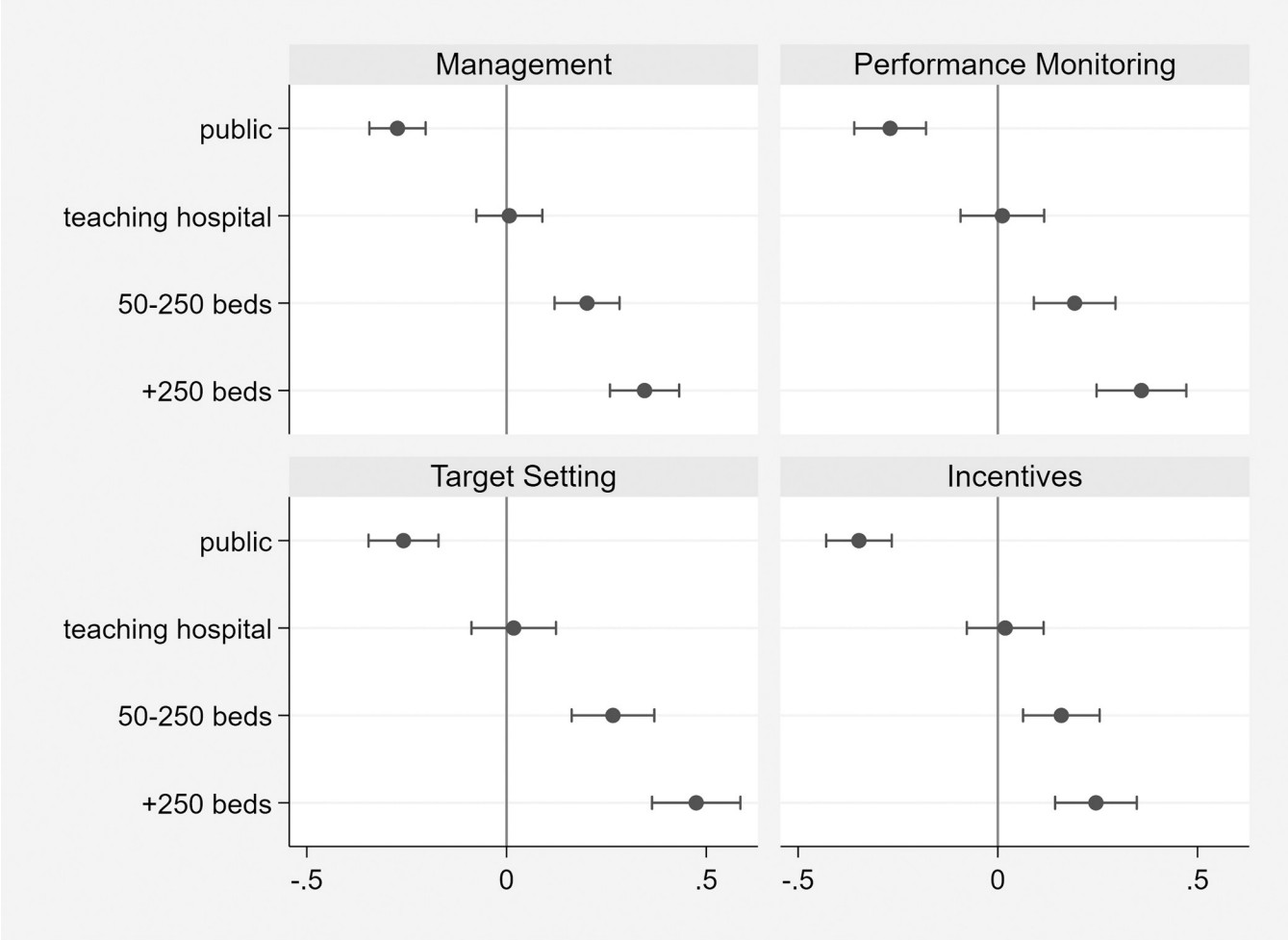

**Fig 3. Differences in management dimensions.** Source: World Management Survey (Hospitals sub-sample).

always associated with better management practices. This is in line with the evidence suggesting that smaller hospitals can rely more on informal relations, supervision by managers is easier, and organizational procedures and routines are more simple. With reference to a small sized hospital (less than 50 beds), we find that a medium (50 to 250 beds) and a large hospital (more than 250 beds) are associated with a score in management practices that is about 6% to 10% higher, respectively (i.e. the latter correspond to a half-standard deviation in scores). The highest increase in management scores with hospital size is associated with 'target setting', supporting the view that a larger organization commands a clear strategy in the planning and definition of objectives. Similar results are obtained when using the log of the number of hospital beds as a proxy for hospital size (see S3bis Table in S1 Appendix).

In a further exercise, we also interact the public sector dummy with the hospital size dummies. Results show that the negative gap in management practices associated to public sector hospitals is much larger in large hospitals–the gap due to public ownership is almost doubled (20%)–, while there are no statistically significant differences between small and medium sized hospitals. In other words, those factors that seem to reduce the performance of managers in public owned hospitals appear to be reinforced in larger organization, to the point that almost half of the higher management score associated to larger hospitals is dissipated by public ownership.

In the above analysis, we have pooled all countries together relegating any cross-country difference in management practices to the country specific fixed-effects. However, since the structure of the healthcare system, the organization and functioning of hospitals, and the role of the public sector differ substantially across countries, the relationship between management practices and hospital characteristics could also be affected in different ways. To evaluate the relevance of the heterogeneous effects across countries, we estimate our model separately by country (see also S4 Table in A1 Apepndix). The lower management score for public sector hospitals is generally confirmed in all countries, except for Canada and Germany, where it is not statistically significant. Also the positive gradient in management scores with hospital size is found in most countries, particularly in larger hospital where management scores are significantly higher (except in France where most of the hospitals in the sample have more than 250 beds).

## Robustness checks

In this section, we test the robustness of the effect of public ownership on hospitals' management score using alternative specifications or different empirical strategies. Results of the different exercises are reported in the Supporting information section. For example, when the model is estimated with the log of the number of beds as a proxy for hospital size instead of categorical dummies for small, medium and large hospital the results are unchanged (see S3bis Table in S1 Appendix). In S5 Table in S1 Appendix, we report the gross effect of hospital's public ownership including only country fixed-effects (column 1); then, we add to the specification additional controls, such as observable hospital characteristics (column 2). Next, we estimate the propensity score of public ownership and use the score distribution (i.e. deciles) to stratify the sample and derive the ATE (Average Treatment Effect) (column 3).

Finally, we present the results of a two-step Heckman selection model, using a dummy for the presence of competitors in the local market as instrument for hospital's public ownership. The result of the selection equation shows, as expected, that the presence of competitors is negatively correlated with a public ownership. The correlation between the unobservables of the selection equation and the management practices equation is positive though not statistically significant, suggesting that the negative management gap estimated in the previous sections is unlikely to be driven by selection in the unobservables.

## Discussion

Much of the previous analysis has shown that larger hospitals are better managed, and that publicly owned hospitals tend to underperform compared to private ones. Notice however, as previously discussed, that some care should be used in interpreting the results, since large hospitals may well have better procedures, but it could also be the case that better managers are more likely to be hired in larger organization. Other characteristics may also be relevant to explain the observed dispersion in management practices, for example the type of hospital, whether general, devoted to training or to research activity), the characteristic of the served population and the overall organizational complexity [14,15]. These different mechanisms, due to the lack of information, cannot be clearly disentangled in our setting. Moreover, while the difference in management scores between public and private hospitals does not seem to be driven by unobservable characteristics associated to hospital ownership, such patterns raise the question of what other factors could explain management practices dispersion. Several studies have explored the factors that are associated with higher managerial standards, such as: higher-rated hospital boards [25], proximity to medical schools and MBA courses [19], hospital quality and patients' choices [26], and hospitals' competition [3].

The effect of having more competitors could also differ between public and private hospitals, for example the discipline effect of competition on management standards may depend on the substitutability of services provided by hospitals, on their proximity, size and other factors. While many of these factors have been analyzed in the literature, here we focus attention on the hypothesis that better management practices are associated with increased competition in the local market, disentangling the effect between public and private hospitals. We use the information reported in WMS data on the degree of competition that hospitals face in the local market, as reported by the respondent. The average number of (hospital) competitors in the public sector is 3.1, while in the private sector is 6.5. In Fig 4, we illustrate the (unconditional) relationship between the management practices score and hospital competition in public and private hospitals. In privately owned hospitals we find no statistically significant correlation, while evidence for public hospitals shows that reduced competition is associated with worse management practices. A simple regression between management scores and the number of competitors, controlling for hospital attributes and country fixed-effects, returns a coefficient of 0.003 (pvalue = 0.003) and 0.011 (p-value = 0.004)–with an elasticity of 0.006 and 0.013 –for private and public hospitals, respectively. To get a sense of the magnitude of the effect of competition, the above estimates indicate that an increase of 5 (hospital) competitors–

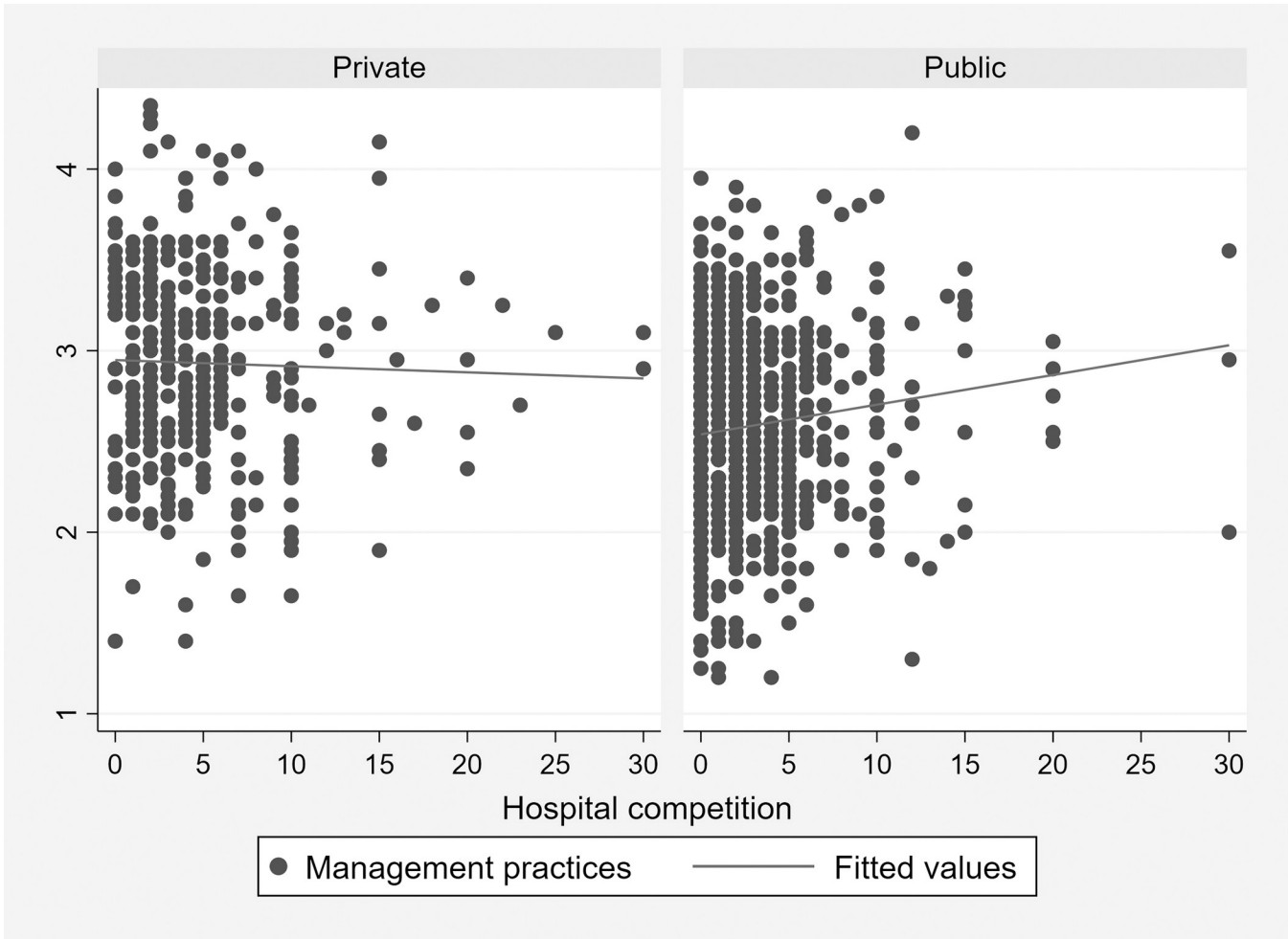

**Fig 4. Management practices and hospital competition, public and private.** Source: World Management Survey (Hospitals sub-sample).

comparable to a shift from the first to the third quartile of the competitors' distribution–is associated with an increase in the managerial score of private (public) hospitals equal to 5% (15%) of a standard deviation. Notice however, that since better management may reduce the probability that other hospitals enter the market, reverse causality may induce negative selection and result in an underestimation of the (positive) effect of competition. Bloom et al. [2] exploit a natural experiment in hospital closure and find a larger impact.

## Conclusions

Using data from the WMS initiative, we have documented the existence of significant variation in management practices in the healthcare sector both between countries and, within countries, across hospitals in terms of size and public-private ownership. In particular, while larger hospitals have been found to be better managed compared to small ones, managerial practices in publicly owned hospitals do show a lower score compared with private ones. In particular, public sector hospitals tend to underperform in the provision of incentive schemes to promote and reward highly motivated employees, or remove poor performers. Other reasons discussed in the literature to account for the heterogeneity in management practices across hospitals are: the vocation of the hospital—whether general, devoted to training or to research—, the characteristic of the served population and the overall organizational complexity. Anecdotal evidence from case studies also suggests that in public sector hospitals, promotion is often based on seniority rather than on merit, performance-related-pay is rarely used, while strong union representation introduces limits and rigidities in human resources management.

Notice that, while the lack of information and the non-experimental approach used in this study somewhat limit the latitude of our results, still a number of original contributions emerge from the empirical analysis. First, we show that public owned hospital are characterized by a lower overall management score: public ownership, on average, implies a reduction of about 10% in management scores. We also report evidence consistent with the fact that the observed difference in management scores, between public and private hospitals, is not correlated with unobservable hospital characteristics associated with ownership. Second, when we analyze the single components of management practices, we find a similar magnitude in terms of monitoring and target setting practices, while in terms of incentives–i.e. to recruit, retain, promote and reward employees—a much larger gap in management scores (approximately 30%) is found. Third, we find no statistically significant differences in management practices between teaching hospitals and other hospitals, which may conceal contrasting effects between higher skills but more complexity in performance targets. Fourth, we show that larger hospitals, conditional on ownership, are always associated with better management practices, suggesting that organizational complexities, supervision and regulations command higher management scores. In practice, we find that a large hospital (more than 250 beds), compared with a small hospital (less than 50 beds), is associated with a score in management practices that is about 10% higher. Fifth, in line with the literature showing that management practices also depend on cultural and environmental factors, we find some heterogeneity in the effect of public hospital ownership and hospital size, on management practices, across countries. Finally, we report casual evidence suggesting that higher management scores are associated with increased competition in the local market, particularly for public hospital, while such relationship is not statistically significant for private hospitals.

With all the caveats previously discussed, this study can contribute to the literature on the determinants of management practices with new empirical evidence on both the measurement of management scores, as well as on the differences in the distribution of these scores between public and private hospitals. While particular care should be used in drawing policy

implications for health policies, our results support previous findings suggesting that better managerial skills translate in higher productivity, better healthcare quality and improved patients' health.

## Supporting information

**S1 Appendix.**
(DOCX)

**S1 File. Data & codes.**
(ZIP)

## Acknowledgments

I am grateful to Tito Boeri, Daniele Checchi, Alessandra Fenizia, Gilberto Turati and participants at the FRDB Conference on "Public Sector Jobs" (Bocconi University, October 2020) and Bank of Italy–World Bank Workshop on "Measuring and Evaluating Public administration Efficiency (Rome, September 2021) for their comments. I am particularly grateful to the editor and two anonymous referees for their constructive comments. I would like to express my gratitude to Nick Bloom, Renata Freitas, John Van Reenen, Raffaella Sadun, Daniela Scur and Thomaz Teodorovicz for sharing additional information not available in the public-use database.

## Author Contributions

**Writing – original draft:** Claudio Lucifora.

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
