## [Decision Letter · Decision Letter 0]

4 Jun 2022

PONE-D-22-07840Management Practices in Hospitals: A Public-Private ComparisonPLOS ONE

Dear Dr. lucifora,

Thank you for submitting your manuscript to PLOS ONE. After careful consideration, we feel that it has merit but does not fully meet PLOS ONE’s publication criteria as it currently stands. Therefore, we invite you to submit a revised version of the manuscript that addresses the points raised during the review process.Please find below other specific comments on the paper.

We look forward to receiving your revised manuscript.

Kind regards,

Anna Prenestini, Ph.D.

Academic Editor

PLOS ONE

Journal Requirements:

2.  Please include a caption for figure 4.

3. Please upload a copy of Figure 6, to which you refer in your text on page 8. If the figure is no longer to be included as part of the submission please remove all reference to it within the text

Additional Editor Comments :

Dear Authors, thank you for submitting the paper to PLOS ONE.

The topic is really interesting and useful for the progress of healthcare management studies.

However, the expert reviewers raised several concerns which prevent from publication on PLOS ONE in its current form, especially for a lack of an adequate literature review on public administration and management field, and on the specific characteristics of healthcare organizations and management. Moreover, there is a common request to reorganize the structure of the paper, along with several other comments.

My advice is to follow carefully the requests of the two reviewers and submit the paper for a second round of review after these improvements.

Reviewers' comments:

Reviewer's Responses to Questions

**Comments to the Author**

1. Is the manuscript technically sound, and do the data support the conclusions?

Reviewer #1: Partly

Reviewer #2: Yes

2. Has the statistical analysis been performed appropriately and rigorously? 

Reviewer #1: Yes

Reviewer #2: Yes

3. Have the authors made all data underlying the findings in their manuscript fully available?

Reviewer #1: No

Reviewer #2: Yes

4. Is the manuscript presented in an intelligible fashion and written in standard English?

Reviewer #1: Yes

Reviewer #2: Yes

5. Review Comments to the Author

Reviewer #1: Management Practices in Hospitals: A Public-Private Comparison by Claudio Lucifora

Referee Report

The paper addresses a salient issue in health policy and management and has potential intriguing implications. It is also clearly written and easy to follow. However, it has several crucial weaknesses that need to be addressed by the author before becoming publishable. I will present them according to the structure of the paper.

Abstract

The first part of the abstract is not relevant as it is already known that management matters in emergency situation and the paper is based on rather old data. I suggest to radically change the first three lines of the abstract. I would openly state when data were collected as in the last decade there has been major changes in healthcare management in several countries, including Italy. Finally, I find the final part too simplistic and with no reference to the literature about how to improve management in the public sector. In this respect I suggest the author to get a bit more familiar with the public and healthcare management literature. The references show that he is not aware of a scholarly debate in the field of Public Administration and Public Management.

2. Management practices in hospitals

• The reference Chandra and Staiger (2007) appears unrelated to the statement made by the authors by the similar incentive structure in public and private hospital. Indeed, these two types of organizations tend to have different institutional goals; as it is well known private hospitals are mainly dominated by the search of shareholder value while public organizations are expected to pursue the public interest and they are, by definition, directly or indirectly, accountable to politics (see for example the Oxford Handbook on Public Management for an overview (Ferlie et al., 2005). Also, it is unclear how the survey and the authors treated non-profit hospitals, that are organizations that are generally owned by private entities but cannot share profits. It should specified this distinction and, if possible, it should be taken into account in the analysis.

• This sentence at the end of the section: “It is important to stress that what is under investigation here is not the quality of healthcare or the services provided to patients, nor the resilience of hospitals to the coronavirus outbreak, but simply the management practices adopted in public and private hospitals, during normal times, to monitor performance, set targets, and recruit, retain and motivate the personnel. Is very important,” is crucial. Particularly, it is essential that the author explains why he chose these three areas among all possible avoidable dimensions collected in the interviews, including some that are healthcare management specific.

• It should be reported when the data were collected given changes occurring in healthcare system worldwide

• It should reported (and justified) that the survey is based on interviews with clinicians and not managers strictu senso. As far as I understand, they acted as key informant about what happened in their hospitals. Obviously, they reported what perceived through the lenses of the clinical profession (see Numerato et al. 2012) to get a first idea about the relationship between management and medical doctors. It may be also useful to read a very recent paper that, by chance, investigated the same professionals in two Italian regions (Fattore et al, 2022).

• Please provide in the main text more info about the interview. If I remember well they are presented in one of the papers authored by Bloom.

3. Data and descriptive statistics

My preference is for papers that follow a traditional format: introduction, literature review (basically lacking in this paper), methods (including data and statistical methods), results, discussion and conclusions. It facilitates reading the paper.

• Present the statistical analysis of the difference in scores between public and private hospitals

• While Italy is missing in the figure presenting data country by country?

• What’s the meaning of quoting Syverson 2011? Doesn’t this paper refer to management in general? Anyway, it should be explained better why the parallel between the two papers matter

4. Main results

See my comment above about the structure of the paper. The regression model, with its various specifications, is the heart of the paper. The model, although simple, makes sense and it is clearly presented. Still needs some improvements (part in the methods section and part in the result section).

• I would avoid to use the expression management quality as the three indicators measure a part of what management is or can be

• I would avoid the expression management styles because it has a specific and narrower meaning in management studies

• Why the size of the hospitals is defined with two dummies (three classes). Don’t you have the stated size of the hospitals so to use it as a continuous variable? Please explain and justify

• This sentence should be moved in the discussion section where limitations of the study should be openly presented. “Notice that, as previously discussed, some care should be used in interpreting the above results, since large hospitals may well have better procedures, but it could also be the case that better managers are more likely to be hired in larger organization. These different hypotheses, however, cannot be disentangled in our setting”.

4.1 Robustness Check

I am not convinced by the Heckman model specification where the presence of competitor is used as IV. As rightly stated, the exclusion restriction assumption is unlikely to be respected. Why the level of competition is not simply used as control variable? In this respect, how competition is measured? Anyway, the part of the model presented in the discussion needs to presented earlier (in the method and results sections). More info are also needed to understand the variables to characterize hospitals; which are they? Are data reported by the key informants or collected from institutional sources (e.g., official eb sites)?

4.2 Discussion

This and the following sections are very scant with too few references and over simplifications. In the following I present some points of reflections.

• Larger hospitals are more difficult to be managed and thus require, in addition to others, the three elements considered by the authors. Smaller hospitals can perform well even with more basic management systems because informal relations, direct supervision by CEO and other top managers is easier, and organizational procedures and routines are more simple

• A large literature show that private and public management differences are substantial. Is it useful to use the same dimensions to measure management quality in the public and private sector? This was for sure the wish of the extreme New Public Management literature (see for example (Osborne and Gabler, 1992). For a critique to NPM see (Hood and Dixon, 2015). Maybe, looking for a co-author in the field of management and/or specifically in public management could make the difference, because she/he could put the empirical results of the study in the cultural context of disciplines that are debating these issues for decades.

Overall, this paper is potentially interesting and covers issues where there are relevant empirical gaps. However, as it stands is not publishable by PLOSE ONE. It needs a radical revision.

I do hope that my suggestions can help improve the paper.

References

Fattore G, Numerato D, Salvatore D. Do Policies affect Management? Evidencce from a survey of clinicians in the Italian NHS. Health Service Management Research. 2022.

Ferlie E, Lynn L, Pollitt C edr. The Oxford Handbook of Public Management. Oxford University Press. 2007.

Hood C, Dixon R. A Government that worked better and costs less? Oxford University Press. 2015.

Numerato D, Salvatore D, Fattore G. The impact of management on medical professionalism: A review. Sociology of Health and Illness 2012; 34 (4): 626-644.

Osborne D, Gaebler T. Reinventing Government. New York. Penguin Press. 1992

Reviewer #2: Overall comments

The paper explores how management practices in healthcare sector varies across seven OECD countries according to hospital size, public-private ownership and level of competition. The paper is consistent with the contemporary shared interest in the study of healthcare management and the topic pertains to scope and aims of the journal. Author’ contributions can be relevant because the study employs a large hospitals sample covering seven OECD countries and it properly employs the chosen methodologies. Therefore, I generally found the article interesting and with a good potential to contribute to the literature. Nevertheless, there is still some work to be done as several major issues strongly prevent the paper to be published. The following part of the review will provide some specific comments about each section of the paper.

1. Introduction

I would suggest to deeply revise and re-articulate the introduction. Indeed, while in the first paragraph the introduction provides some interesting arguments to motivate the practical relevance of the topic, by suggesting that management practices are key determinants of healthcare performance and efficiency, I would suggest the authors to better elaborate the review of previous studies on the determinants of ‘management practices’, especially for what concerns the ownership structure and identity. This would help the author to identify and develop the related research gap, to motivate the research questions so as to explain how his study could contribute to the literature on the topic. I think this is a major point that might prevent the article’s contribution to the literature and to the journal.

2. Management practices in hospitals

While reading this section my wait was to read a section concerning previous studies on management practices above all in terms of determinants. Actually, the section represents a brief description of items employed to measure “management practices” in the study, as well as to explain that they represent key determinants of performance (that is not the authors’ aim, as declared in the introduction). I would suggest moving the content of this section to the section 3, that I would suggest to entitle “3. Methodology”, to articulate as later discussed. In doing so, I would suggest to add a new section 2 devoted to the literature review on organizational determinants of management practices (especially in terms of ownership structure). As previously stated, I think that the lack of previous studies assessment is a major point that might prevent the article’s contribution to the literature and to the journal.

3. Data and descriptive statistics and 4. Main results

As previously stated, I would suggest to add a new section “3. Methodology” to articulate as follows:

3.1 Management practices in hospitals

This section would include the content of the previous version of section 2

3.2 Data and analysis

This section would include some of the text included in the section “3. Data and descriptive statistics” and section “4. Main results”.

More specifically I would suggest to separate the analysis of the employed methodology from the descriptive and regression results, that I would suggest to include within a new section entitled “4. Findings” articulated as follows: “4.1 Descriptive findings”, “4.2 Regression results”, “4.3 Robustness Checks”.

While section 3 (and related subsections) should describe the sample, the method, variables and analysis, section 4 (and related subsections) should describe descriptive findings, results coming from more advanced statistical analysis and robustness checks.

Finally, still concerning section 3 and specifically with regard to the regression analysis, I would suggest enlarging the list of control variables in order to include more controls that could influence the reported findings. For example, I would suggest the authors to control for (i) type of hospitals (general hospitals, teaching hospitals, research hospitals); (ii) hospital performance (e.g. quality and efficiency) (as it might possible that the adoption of management practices is more likely to occur within better performers); (iii) characteristics of served population; (iv) hospital complexity.

4.2 Discussion

I would suggest separating the discussion from the findings within a new section “5. Discussion”. I also think that actually the section is a little bit rare and this is due to the scarce assessment of previous studies on the literature background on the topic.

5. Concluding Remarks

I would suggest the authors to better reflect about the academic implications. Moreover, they could better explain in which way their article provides contribution to the theory. Authors should also reflect on the limitations of the study and provide them.

I hope that these comments can help the authors to better develop their study and to bring out its full potential. I wish them good luck with their research!

6. PLOS authors have the option to publish the peer review history of their article (what does this mean?). If published, this will include your full peer review and any attached files.

Reviewer #1: **Yes: **Giovanni Fattore

Reviewer #2: No

---

## [Author Response · Author response to Decision Letter 0]

17 Jul 2022

Reviewers' comments:

Reviewer's Responses to Questions

Reviewer #1: No

Reviewer #2: Yes

ANSWER: The questionnaires and the original dataset are fully available on the World Management Survey public repository (https://worldmanagementsurvey.org/ ). The data used in the paper have been downloaded on the 20th December 2019. This reported in the data section and in the aknowledgements. Also the Stata codes for the empirical analysis have been made available as .zip file in the Supplementary information.

Reviewer #1: Management Practices in Hospitals: A Public-Private Comparison by Claudio Lucifora

Referee Report

The paper addresses a salient issue in health policy and management and has potential intriguing implications. It is also clearly written and easy to follow. However, it has several crucial weaknesses that need to be addressed by the author before becoming publishable. I will present them according to the structure of the paper.

ANSWER: thank you very much for the useful comments and suggestions, which helped me to revise and substantially improve the paper. All the issues raised by yourself and by another referee have been implemented in the new draft resubmitted to PLOS ONE. In particular, the paper has been thoroughly reorganized, both adding a review of the literature and new references, as well as further discussion of the results hopefully improving the clarity of the paper. In what follows, I provide a detailed answer to the points you raised (for your convenience, they are reported below in italics). I really hope that this revision answers all your main concerns.

1. Abstract

The first part of the abstract is not relevant as it is already known that management matters in emergency situation and the paper is based on rather old data. I suggest to radically change the first three lines of the abstract. I would openly state when data were collected as in the last decade there has been major changes in healthcare management in several countries, including Italy. Finally, I find the final part too simplistic and with no reference to the literature about how to improve management in the public sector. In this respect I suggest the author to get a bit more familiar with the public and healthcare management literature. The references show that he is not aware of a scholarly debate in the field of Public Administration and Public Management.

ANSWER: The abstract has been thoroughly revised. I openly state the type of data used and when they were collected. The main contributions of the paper are summarized, while comments related to the policy implications have been relegated to the concluding section. I have added a new section with a review of the literature on management practices in the public sector, and a specific focus on healthcare. The list of references has been updated to include contributions from the field of Public management.

2. Management practices in hospitals

• The reference Chandra and Staiger (2007) appears unrelated to the statement made by the authors by the similar incentive structure in public and private hospital. Indeed, these two types of organizations tend to have different institutional goals; as it is well known private hospitals are mainly dominated by the search of shareholder value while public organizations are expected to pursue the public interest and they are, by definition, directly or indirectly, accountable to politics (see for example the Oxford Handbook on Public Management for an overview (Ferlie et al., 2005). Also, it is unclear how the survey and the authors treated non-profit hospitals, that are organizations that are generally owned by private entities but cannot share profits. It should specified this distinction and, if possible, it should be taken into account in the analysis.

ANSWER: The reference to Chandra and Staiger has been removed. A more thorough discussion of the different goals and of the direct and indirect effect of contextual features differentiating private and public hospitals has been added to the literature review section. The role of politics both in terms of accountability and presence in the board of large public hospitals has been also discussed. 

• This sentence at the end of the section: “It is important to stress that what is under investigation here is not the quality of healthcare or the services provided to patients, nor the resilience of hospitals to the coronavirus outbreak, but simply the management practices adopted in public and private hospitals, during normal times, to monitor performance, set targets, and recruit, retain and motivate the personnel. Is very important,” is crucial. Particularly, it is essential that the author explains why he chose these three areas among all possible avoidable dimensions collected in the interviews, including some that are healthcare management specific.

ANSWER: Thank you for raising this point, which gave me the opportunity to explain better how these three dimensions were computed out of the 18 simple indicators collected in the interviews. Reference to Table S1 in the Supporting information has also been made.

• It should be reported when the data were collected given changes occurring in healthcare system worldwide

ANSWER: The details of data collection within the “World Management Survey” initiative such as, time span, countries covered, n. of hospitals have been reported in the Abstract, in the Introduction and in the Data section.

• It should reported (and justified) that the survey is based on interviews with clinicians and not managers strictu senso. As far as I understand, they acted as key informant about what happened in their hospitals. Obviously, they reported what perceived through the lenses of the clinical profession (see Numerato et al. 2012) to get a first idea about the relationship between management and medical doctors. It may be also useful to read a very recent paper that, by chance, investigated the same professionals in two Italian regions (Fattore et al, 2022).

ANSWER: Thank you for suggesting these useful references I was not aware of. The fact that the interviews were conducted with clinicians and not with managers, as well as the issue of perception of managerial practices is now discussed in the methodological and data sections, highlighting both the pros and cons.

• Please provide in the main text more info about the interview. If I remember well they are presented in one of the papers authored by Bloom.

ANSWER: The details of the double-blind methodology used in the WMS initiative has been thoroughly described in the methodological section.

3. Data and descriptive statistics

My preference is for papers that follow a traditional format: introduction, literature review (basically lacking in this paper), methods (including data and statistical methods), results, discussion and conclusions. It facilitates reading the paper.

ANSWER: This is exactly how the current version of the paper is organized

• Present the statistical analysis of the difference in scores between public and private hospitals

ANSWER: This is done in Fig 1

• While Italy is missing in the figure presenting data country by country?

ANSWER: Italy is included, only Sweden is missing in Fig 2, and a justification is provided

• What’s the meaning of quoting Syverson 2011? Doesn’t this paper refer to management in general? Anyway, it should be explained better why the parallel between the two papers matter

ANSWER: It is true that Syverson 2011 refers to management in general, the idea was to provide further evidence in support to the scoring methodology used in the WMS initiative, showing that it is consistent with observed cross-country differences in productivity. Hopefully, this is now explained more clearly.

4. Main results

See my comment above about the structure of the paper. The regression model, with its various specifications, is the heart of the paper. The model, although simple, makes sense and it is clearly presented. Still needs some improvements (part in the methods section and part in the result section).

• I would avoid to use the expression management quality as the three indicators measure a part of what management is or can be

ANSWER: Thank you for these detailed comments and suggestions. I have added a sub-section in the ‘Methodology’ section to better explain the empirical model and the different specifications. I have avoided using the expression ‘management quality’ and when necessary I used ‘management standards’ or ‘management scores’.

• I would avoid the expression management styles because it has a specific and narrower meaning in management studies

ANSWER: I have avoided using the expression ‘management styles’ and used ‘management dimensions’ instead.

• Why the size of the hospitals is defined with two dummies (three classes). Don’t you have the stated size of the hospitals so to use it as a continuous variable? Please explain and justify

ANSWER: Thank you for raising this point. Let me first say that hospital size in the model is merely a control variable. Having said this, given that the baseline specification mainly includes dummies, I generally prefer to be consistent and define most controls as dummies. However, given your concerns, I have re-estimated the model with the log of the number of hospital beds (i.e. as a proxy for hospital size) and reported the results in table S3bis in the Supporting information. Reference to the results (essentially unchanged) and the table are given both in the results and the robustness sections.

• This sentence should be moved in the discussion section where limitations of the study should be openly presented. “Notice that, as previously discussed, some care should be used in interpreting the above results, since large hospitals may well have better procedures, but it could also be the case that better managers are more likely to be hired in larger organization. These different hypotheses, however, cannot be disentangled in our setting”.

ANSWER: thank you, I have done it as you suggested

4.1 Robustness Check

I am not convinced by the Heckman model specification where the presence of competitor is used as IV. As rightly stated, the exclusion restriction assumption is unlikely to be respected. Why the level of competition is not simply used as control variable? In this respect, how competition is measured? Anyway, the part of the model presented in the discussion needs to presented earlier (in the method and results sections). More info are also needed to understand the variables to characterize hospitals; which are they? Are data reported by the key informants or collected from institutional sources (e.g., official eb sites)?

ANSWER: Thank you for the comments, I do share some of the criticism of the referee. While I do not want to oversell the results, still I believe that with the due caveats, the Heckman model can provide additional evidence on the sensitivity of the results to alternative specifications and the role of unobservables. Unfortunately, with the available data it is not possible to do more than this. In the revised version I have addressed all the questions raised above, hopefully improving the clarity of the text and providing additional details. For example, the model is now introduced earlier in the methods/empirical strategy section. More information on definition and measurement of hospital characteristics has been given. In practice: (i) competition is defined as the number of hospitals (competitors) that are present in the local market in which the hospital operates, as reported by the subject interviewed; (ii) not external sources of information have been used, as it would have been impossible to match such information at the hospital level.

4.2 Discussion

This and the following sections are very scant with too few references and over simplifications. In the following I present some points of reflections.

• Larger hospitals are more difficult to be managed and thus require, in addition to others, the three elements considered by the authors. Smaller hospitals can perform well even with more basic management systems because informal relations, direct supervision by CEO and other top managers is easier, and organizational procedures and routines are more simple

ANSWER: In the ‘Results’ section the discussion on the role of hospital size has been enriched, both highlighting the importance to control for hospital size, as well as in the interpretation of the results. 

• A large literature show that private and public management differences are substantial. Is it useful to use the same dimensions to measure management quality in the public and private sector? This was for sure the wish of the extreme New Public Management literature (see for example (Osborne and Gabler, 1992). For a critique to NPM see (Hood and Dixon, 2015). Maybe, looking for a co-author in the field of management and/or specifically in public management could make the difference, because she/he could put the empirical results of the study in the cultural context of disciplines that are debating these issues for decades.

ANSWER: I agree that a test of a public-private gap in management standards could appear rather simplistic to an expert of public management given the complexities involved. However, the main contribution of this paper, and of the literature related with the WMS initiative, is exactly to provide a quantification of such differences and attach a statistical significance to it. This is what this paper does. The efforts behind the WMS methodology for data collection and double-blind coding are not trivial and span across a large number of practices, which have been used in the empirical analysis. While the paper still suffer from some weaknesses in explaining the mechanisms that lie behind the estimated lower management scores in public hospitals, the ‘Discussion’ section does provide a number of explanations, with a quantification of the role of hospital competition in the local market on management practices. 

I thank you for suggesting to enrich the co-authorship of the paper with public management skills, but at this stage such step is beyond the scope of the present paper. 

Thank you also for providing a number of references, which have helped me enormously to grasp some of the issues from a different field. They have been all added to the list of references. 

Reviewer #2: Overall comments

The paper explores how management practices in healthcare sector varies across seven OECD countries according to hospital size, public-private ownership and level of competition. The paper is consistent with the contemporary shared interest in the study of healthcare management and the topic pertains to scope and aims of the journal. Author’ contributions can be relevant because the study employs a large hospitals sample covering seven OECD countries and it properly employs the chosen methodologies. Therefore, I generally found the article interesting and with a good potential to contribute to the literature. Nevertheless, there is still some work to be done as several major issues strongly prevent the paper to be published. The following part of the review will provide some specific comments about each section of the paper.

ANSWER: thank you very much for the useful comments and suggestions. The paper has been thoroughly revised following the comments raised by yourself and by another referee. In particular, the paper has been reorganized, adding sections to provide additional information on literature and better focusing the contribution to the existing literature.

In what follows I provide a detailed answer to the points you raised (for your convenience, they are reported below in italics). I really hope that this revision answers all your main concerns.

1. Introduction

I would suggest to deeply revise and re-articulate the introduction. Indeed, while in the first paragraph the introduction provides some interesting arguments to motivate the practical relevance of the topic, by suggesting that management practices are key determinants of healthcare performance and efficiency, I would suggest the authors to better elaborate the review of previous studies on the determinants of ‘management practices’, especially for what concerns the ownership structure and identity. This would help the author to identify and develop the related research gap, to motivate the research questions so as to explain how his study could contribute to the literature on the topic. I think this is a major point that might prevent the article’s contribution to the literature and to the journal.

ANSWER: thank you very much for these comments and the suggestion to better focus the discussion on management practices, in terms of ownership and identity. To address this point, which was also raised by another referee, I have enriched the discussion on the literature on the determinants of management practices adding a section on the ‘Literature review’, and a specific sub-section on healthcare. Also the list of references has been updated.

2. Management practices in hospitals

While reading this section my wait was to read a section concerning previous studies on management practices above all in terms of determinants. Actually, the section represents a brief description of items employed to measure “management practices” in the study, as well as to explain that they represent key determinants of performance (that is not the authors’ aim, as declared in the introduction). I would suggest moving the content of this section to the section 3, that I would suggest to entitle “3. Methodology”, to articulate as later discussed. In doing so, I would suggest to add a new section 2 devoted to the literature review on organizational determinants of management practices (especially in terms of ownership structure). As previously stated, I think that the lack of previous studies assessment is a major point that might prevent the article’s contribution to the literature and to the journal.

ANSWER: I agree that the main focus of the paper is the relationship between hospital ownership and management practices, and not on performance. In the revised version, I have clarified that reference to performance is only used to motivate the analysis, while the main focus is on the determinants of management practices. As you recommended, I have added a section with a review of the literature on the determinants of management practices with a specific focus on healthcare and ownership. Most of the literature on the measurement of management practices has been moved to the (new) Methodology section.

3. Data and descriptive statistics and 4. Main results

As previously stated, I would suggest to add a new section “3. Methodology” to articulate as follows:

3.1 Management practices in hospitals

This section would include the content of the previous version of section 2

3.2 Data and analysis

This section would include some of the text included in the section “3. Data and descriptive statistics” and section “4. Main results”.

More specifically I would suggest to separate the analysis of the employed methodology from the descriptive and regression results, that I would suggest to include within a new section entitled “4. Findings” articulated as follows: “4.1 Descriptive findings”, “4.2 Regression results”, “4.3 Robustness Checks”.

While section 3 (and related subsections) should describe the sample, the method, variables and analysis, section 4 (and related subsections) should describe descriptive findings, results coming from more advanced statistical analysis and robustness checks.

ANSWER: Your constructive comments were very useful to re-organize the manuscript. Following your suggestions I have re-organized the sections in ‘Methodology’ and the ‘Main results’. In particular, I have created a ‘Methodology’ section, with the following sub-sections: (i) Measurement of management practices, (ii) Data and analysis. The section with the main findings is now titled ‘Main Results’ with the following sub-sections: (i) Descriptive findings, (ii) Regression Results, (iii) Robustness checks.

Finally, still concerning section 3 and specifically with regard to the regression analysis, I would suggest enlarging the list of control variables in order to include more controls that could influence the reported findings. For example, I would suggest the authors to control for (i) type of hospitals (general hospitals, teaching hospitals, research hospitals); (ii) hospital performance (e.g. quality and efficiency) (as it might possible that the adoption of management practices is more likely to occur within better performers); (iii) characteristics of served population; (iv) hospital complexity.

ANSWER: In the ‘Data and analysis’ sub-section I carefully describe the issue of the restricted number of controls included in the baseline specification. In the description of the methods I discuss the implications that a restricted number of controls could have on the robustness of the estimates, i.e. bias the results due to the omitted variables. In the same sub-section I also discuss the hypothesis of the potentially non-random distribution of public ownership across hospitals. The robustness sub-section tries to address some of the above issues, the main results of these exercises are reported in Table 3Sbis (in the Supporting information appendix), where I report the sensitivity of the public hospital dummy estimates with a different set of controls. The estimated management score gap associated with public hospital is hardly changed under the different specifications. Unfortunately the set of hospital-level controls that can be included is mandated by the information available in the dataset and essentially limited to hospital size (dummies or log number of beds). 

In table 5S of the the Supporting information I report additional robustness checks which leverage on the distribution of the unobservables to provide some additional (indirect) support to the robustness of the results and the issue of omitted variable bias.

4.2 Discussion

I would suggest separating the discussion from the findings within a new section “5. Discussion”. I also think that actually the section is a little bit rare and this is due to the scarce assessment of previous studies on the literature background on the topic.

ANSWER: I revised the ‘Discussion’ sub-section adding a thorough discussion of the limitation of the empirical analysis. In that sub-section I have also recalled some of the mechanisms discussed in the literature to explain management practice dispersion. In the paper, however, the discussion is mainly casted in terms of hospital competition in the local market, distinguishing between public and private hospitals. That despite its simplicity, I hope that this can be considered an original contribution of this paper, as there are not many studies which have investigated this aspect empirically. 

5. Concluding Remarks

I would suggest the authors to better reflect about the academic implications. Moreover, they could better explain in which way their article provides contribution to the theory. Authors should also reflect on the limitations of the study and provide them.

ANSWER: The Conclusions have been thoroughly revised, highlighting the contribution of the paper to the literature and discussing its limitations.

I hope that these comments can help the authors to better develop their study and to bring out its full potential. I wish them good luck with their research!

ANSWER: Thank you for your detailed and constructive comments. I hope that I have been able to fully incorporate them in the new version of the paper, also showing more clearly which are the original contributions.

---

## [Decision Letter · Decision Letter 1]

17 Jan 2023

PONE-D-22-07840R1Management Practices in Hospitals: A Public-Private ComparisonPLOS ONE

Dear Dr. lucifora,

Thank you for submitting your manuscript to PLOS ONE. After careful consideration, we feel that it has merit but does not fully meet PLOS ONE’s publication criteria as it currently stands. Therefore, we invite you to submit a revised version of the manuscript that addresses the points raised during the review process. Please find below the Academic Editor comments. 

We look forward to receiving your revised manuscript.

Kind regards,

Anna Prenestini, Ph.D.

Academic Editor

PLOS ONE

Journal Requirements:

Additional Editor Comments:

Dear authors,

the reviewers' decisions are polarized. Reviewer 1 is completely satisfied with your improvements.

On the contrary, reviewer 2 raises again some concerns about your methodology and, in particular, the lack of use of some suggested control variables.

I read carefully the justifications by the author and some of them convinced me to assign minor revisions.

However, I suggest an effort in order to use at least the typology of hospitals (mainly the difference between teaching and research hospitals and other hospitals) as a control variable. It may be really interesting.

Otherwise, specify why you decided not to use this control variable.

Kind regards.

Reviewers' comments:

Reviewer's Responses to Questions

**Comments to the Author**

1. If the authors have adequately addressed your comments raised in a previous round of review and you feel that this manuscript is now acceptable for publication, you may indicate that here to bypass the “Comments to the Author” section, enter your conflict of interest statement in the “Confidential to Editor” section, and submit your "Accept" recommendation.

Reviewer #1: All comments have been addressed

Reviewer #2: (No Response)

2. Is the manuscript technically sound, and do the data support the conclusions?

Reviewer #1: (No Response)

Reviewer #2: Partly

3. Has the statistical analysis been performed appropriately and rigorously? 

Reviewer #1: (No Response)

Reviewer #2: No

4. Have the authors made all data underlying the findings in their manuscript fully available?

Reviewer #1: (No Response)

Reviewer #2: Yes

5. Is the manuscript presented in an intelligible fashion and written in standard English?

Reviewer #1: (No Response)

Reviewer #2: Yes

6. Review Comments to the Author

Reviewer #1: (No Response)

Reviewer #2: Overall, I think that the structure of the paper and its theoretical positioning is improved as the author(s) have addressed most of my suggestions. Indeed, most of efforts have been done in terms of improvement of the assessment of prior studies on the topic. However, I think that the paper is still lacking in what concerns the methodology as the regression models still suffer of the use of more control variables. This can prevent the strength and the generalizability of the results.

In the previous review I suggested the authors to include more control variables such as (i) type of hospitals (general hospitals, teaching hospitals, research hospitals); (ii) hospital performance (e.g. quality and efficiency) (as it might possible that the adoption of management practices is more likely to occur within better performers); (iii) characteristics of served population; (iv) hospital complexity. Indeed, in the previous draft of the paper, the authors only used the hospital size as control variable. However, in this new version of the paper, hospital size and country dummies remain the only control variables. The authors answered to the comment that they have discussed the possible implication of omitting these variables. I think that the results in this version could be not strong and generalizable.

On the basis of this, I would suggest again the authors to enrich the regression model in order to make their research more robust. I think that the article is interesting and with a good potential, but this issue can strongly prevent the contribution of the paper to the literature.

7. PLOS authors have the option to publish the peer review history of their article (what does this mean?). If published, this will include your full peer review and any attached files.

Reviewer #1: **Yes: **Giovanni Fattore

Reviewer #2: No

---

## [Author Response · Author response to Decision Letter 1]

9 Feb 2023

Response to Editor’s comments

1. reviewer 2 raises again some concerns about your methodology and, in particular, the lack of use of some suggested control variables. I read carefully the justifications by the author and some of them convinced me to assign minor revisions. However, I suggest an effort in order to use at least the typology of hospitals (mainly the difference between teaching and research hospitals and other hospitals) as a control variable. It may be really interesting..

ANSWER: I agree with you and the referee that including hospital type as additional control variable may prove interesting for the empirical analysis. To meet both your and the referee’s concerns, I applied to the WMS initiative asking for information on “hospital type” which was not available in the public-use database. Luckily I was able to obtain the requested info. The paper has therefore been amended accordingly. In the revised version, I discuss the implications of hospital type for public-private differences in management styles. Next, I re-estimate all the specifications including a dummy proxying for “hospital type” (research/teaching versus other).

A few typos and references have been also amended.

Reviewers' comments:

Reviewer #1: All comments have been addressed

Reviewer #2: 

Overall, I think that the structure of the paper and its theoretical positioning is improved as the author(s) have addressed most of my suggestions. Indeed, most of efforts have been done in terms of improvement of the assessment of prior studies on the topic. However, I think that the paper is still lacking in what concerns the methodology as the regression models still suffer of the use of more control variables. This can prevent the strength and the generalizability of the results.

In the previous review I suggested the authors to include more control variables such as (i) type of hospitals (general hospitals, teaching hospitals, research hospitals); (ii) hospital performance (e.g. quality and efficiency) (as it might possible that the adoption of management practices is more likely to occur within better performers); (iii) characteristics of served population; (iv) hospital complexity. Indeed, in the previous draft of the paper, the authors only used the hospital size as control variable. However, in this new version of the paper, hospital size and country dummies remain the only control variables. The authors answered to the comment that they have discussed the possible implication of omitting these variables. I think that the results in this version could be not strong and generalizable.

On the basis of this, I would suggest again the authors to enrich the regression model in order to make their research more robust. I think that the article is interesting and with a good potential, but this issue can strongly prevent the contribution of the paper to the literature.

ANSWER: I agree with the referee that including additional variable to control for hospital type may prove interesting for the empirical analysis. Unfortunately, such information was not available in the WMS public-use database, and in my previous revision I could not address the referee’s request to add further control variables to the empirical analysis. However, given that the referee considers the “type” of hospital as a serious confounding factor, I approached the WMS initiative and asked for the additional information on hospital type to be provided. Luckily, the information on hospital teaching status was available in the original questionnaire and my request was approved by the WMS initiative. The revised version of the paper, which I am herein submitting, contains an additional variable capturing whether the hospital is a “research/teaching” versus other. Unfortunately, no information on hospital performance was asked in the survey. In practice the paper has been thoroughly revised as follows: first, a discussion of the implications of hospital type for the analysis has been added in the Introduction and in the data sections; second, the empirical analysis has been amended to include a dummy for teaching hospital status in all estimated specifications. The tables and figures have all be revised accordingly. A few typos and references have been also amended.

---

## [Editor Report · Decision Letter 2]

14 Feb 2023

Management Practices in Hospitals: A Public-Private Comparison

PONE-D-22-07840R2

Dear Dr. lucifora,

We’re pleased to inform you that your manuscript has been judged scientifically suitable for publication and will be formally accepted for publication once it meets all outstanding technical requirements.

Kind regards,

Anna Prenestini, Ph.D.

Academic Editor

PLOS ONE

Additional Editor Comments (optional):

Congratulations on the improvement of the paper. Now it is ready for publication on PLOS One.
---

## [Editor Report · Acceptance letter]

16 Feb 2023

PONE-D-22-07840R2 

Management Practices in Hospitals: A Public-Private Comparison 

Dear Dr. Lucifora:

I'm pleased to inform you that your manuscript has been deemed suitable for publication in PLOS ONE. Congratulations! Your manuscript is now with our production department. 

Kind regards, 

on behalf of

Professor Anna Prenestini 

Academic Editor

PLOS ONE